# RoboCLIP:
# One Demonstration is Enough to Learn Robot Policies

Sumedh A Sontakke[1][*], Jesse Zhang[1], Sébastien M. R. Arnold[4],
Karl Pertsch[2,3], Erdem Bıyık[1,2], Dorsa Sadigh[3], Chelsea Finn[3], and Laurent Itti[1]

[1]Thomas Lord Department of Computer Science, University of Southern California
[2]University of California, Berkeley
[3]Stanford University
[4]Google Research

## Abstract

Reward specification is a notoriously difficult problem in reinforcement learning, requiring extensive expert supervision to design robust reward functions. Imitation learning (IL) methods attempt to circumvent these problems by utilizing expert demonstrations but typically require a large number of in-domain expert demonstrations. Inspired by advances in the field of Video-and-Language Models (VLMs), we present RoboCLIP, an online imitation learning method that uses a single demonstration (overcoming the large data requirement) in the form of a video demonstration or a textual description of the task to generate rewards without manual reward function design. Additionally, RoboCLIP can also utilize out-of-domain demonstrations, like videos of humans solving the task for reward generation, circumventing the need to have the same demonstration and deployment domains. RoboCLIP utilizes pretrained VLMs without any finetuning for reward generation. Reinforcement learning agents trained with RoboCLIP rewards demonstrate 2-3 times higher zero-shot performance than competing imitation learning methods on downstream robot manipulation tasks, doing so using only one video/text demonstration. Visit our website for experiment videos.

## 1 Introduction

Sequential decision-making problems typically require significant human supervision and data. In the context of online reinforcement learning [Sutton and Barto, 2018], this manifests in the design of good reward functions that map transitions to scalar rewards [Amodei et al., 2016, Hadfield-Menell et al., 2017]. Extant approaches to manual reward function definition are not very principled and defining rewards for complex long-horizon problems is often an art requiring significant human expertise. Additionally, evaluating reward functions often requires knowledge of the true state of the environment. For example, imagine a simple scenario where the agent must learn to lift an object off the ground. Here, a reward useful for task success would be proportional to the height of the object from the ground — a quantity non-trivial to obtain without full state information. Thus, significant effort has been expounded in developing methods that can learn reward functions either explicitly or implicitly from demonstrations, i.e., imitation learning [Pomerleau, 1988, Ng and Russell, 2000, Abbeel and Ng, 2004, Ziebart et al., 2008]. With these methods, agent policies can either be directly extracted from the demonstrations or trained to optimize rewards functions learned from them.

Imitation learning (IL), however, only somewhat alleviates the need for expert human intervention. First, instead of designing complex reward functions, expert supervision is needed to collect massive

---

[*]corresponding author: ssontakk@usc.edu

37th Conference on Neural Information Processing Systems (NeurIPS 2023).

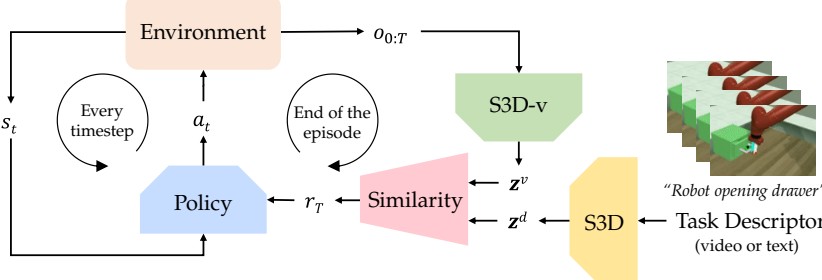

**Figure 1: RoboCLIP Overview.** A Pretrained Video-and-Language Model is used to generate rewards via the similarity score between the encoding of an episode of interaction of an agent in its environment, $\mathbf{z}^v$ with the encoding of a task specifier $\mathbf{z}^d$ such as a textual description of the task or a video demonstrating a successful trajectory. The similarity score between the latent vectors is provided as reward to the agent.

datasets such as RT-1 [Brohan et al., 2022], Bridge Dataset [Ebert et al., 2021], D4RL [Fu et al., 2020], or Robonet [Dasari et al., 2019]. The performance of imitation learning algorithms and their ability to generalize hinges on the coverage and size of data [Kumar et al., 2019, 2022], making the collection of large datasets imperative. Second and most importantly, the interface for collecting demonstrations for IL is tedious, requiring expert robot operators to collect thousands of demonstrations. On the contrary, a more intuitive way to define rewards would be in the form of a textual description (e.g., "*robot grasping object*"), or in the form of a naturalistic video demonstration of the task performed by a human actor in an environment separate from the robotic environment. For example, demonstrating to a robot how to open a cabinet door in one's own kitchen is more naturalistic than collecting many thousands of trajectories via teleoperation in the target robotic environment.

Thus, there exists an unmet need for IL algorithms that 1) require very few demonstrations and 2) allow for a natural interface for providing these demonstrations. For instance, algorithms that can effectively learn from language instructions or human demonstrations without the need for full environment state information. Our key insight is that by leveraging Video-and-Language Models (VLMs)—which are already pretrained on large amount of video demonstration and language pairs— we do not need to rely on large-scale and in-domain datasets. Instead, by harnessing the power of VLM embeddings, we treat the mismatch between a single *instruction's* embedding (provided as a language command or a video demonstration) and the embedding of the video of the current policy's rollout as a proxy reward that will guide the policy towards the desired *instruction*.

To this end, we present RoboCLIP, an imitation learning algorithm that learns and optimizes a reward function based on a single language or video demonstration. The backbone model used in RoboCLIP is S3D [Xie et al., 2018] trained on the Howto100M dataset [Miech et al., 2019], which consists of short clips of humans performing activities with textual descriptions of the activities. These videos typically consist of a variety of camera angles, actors, lighting conditions, and backgrounds. We hypothesize that VLMs trained on such diverse videos are invariant to these extraneous factors and generate an actor-agnostic semantically-meaningful representation for a video, allowing them to generalize to unseen robotic environments.

We present an overview of RoboCLIP in Figure 1. RoboCLIP computes a similarity score between videos of online agent experience with a task descriptor, i.e., a text description of the task or a single human demonstration video, to generate trajectory-level rewards to train the agent. We evaluate RoboCLIP on the Metaworld Environment suite [Yu et al., 2020] and on the Franka Kitchen Environment [Gupta et al., 2019], and find that policies obtained by pretraining on the RoboCLIP reward result in $2 - 3\times$ higher zero-shot task success in comparison to state-of-the-art imitation learning baselines. Additionally, these rewards require no experts for specification and can be generated using naturalistic definitions like natural language task descriptions and human demonstrations.

## 2 Related Work

**Learning from Human Feedback.** Learning from demonstrations is a long-studied problem that attempts to learn a policy from a dataset of expert demonstrations. Imitation learning (IL) methods, such as those based on behavioral cloning [Pomerleau, 1988], formulate the problem as a supervised learning over state-action pairs and typically rely on large datasets of expert-collected trajectories directly demonstrating how to perform the target task [Brohan et al., 2022, Lynch et al., 2022].

However, these large demonstration datasets are often expensive to collect. Another IL strategy is *inverse* RL, i.e., directly learning a reward function from the demonstrations [Ng and Russell, 2000, Abbeel and Ng, 2004, Ziebart et al., 2008, Finn et al., 2016]. Inverse RL algorithms are typically difficult to apply when state and action spaces are high-dimensional. Methods such as GAIL [Ho and Ermon, 2016], AIRL [Fu et al., 2017], or VICE [Fu et al., 2018] partially address these issues by assigning rewards which are proportional to the probability of a given state being from the demonstration set or a valid goal state as estimated by a learned discriminator network. However these discriminator networks still require many demonstrations or goal states to train to effectively distinguish between states from agent-collected experience and demonstration or goal states. On the other hand, RoboCLIP's use of pretrained video-and-language models allows us to train agents that learn to perform target tasks with just *one demonstration* in the form of a video or a language description. Other works instead use human feedback in the form of pairwise comparisons or rankings to learn preference reward functions [Christiano et al., 2023, Sadigh et al., 2017, Biyik et al., 2019, Myers et al., 2021, Bıyık et al., 2022, Brown et al., 2019, Biyik et al., 2020, Lee et al., 2021, Hejna and Sadigh, 2022]. These preferences may require less human effort to obtain than reward functions, e.g., through querying humans to simply rank recent trajectories. Yet individual trajectory preferences convey little information on their own (less than dense reward functions) and therefore humans need to respond to many preference queries for the agent to learn useful reward functions. In contrast, RoboCLIP is able to extract useful rewards from a single demonstration or single language instruction.

**Large Vision and Language Models as Reward Functions.** Kwon et al. [2023] and Hu and Sadigh [2023] propose using large language models (LLMs) for designing and regularizing reward functions that capture human preferences. These works study the reward design problem in text-based games such as negotiations or card games, and thus are not grounded in the physical world. RoboCLIP instead leverages video-and-language models to assess if video demonstrations of robot policies align with an expert demonstration. Prior work has demonstrated that video models can be used as reward functions. For example, Chen et al. [2021] learn a visual reward function using human data and then utilize this reward function for visual model-based control of a robot. However, they require training the reward model on paired human and robot data from the deployment environment. We demonstrate that this paired data assumption can be relaxed by utilizing large-scale vision-language models pretrained on large corpora of human-generated data. The most well-known of these is CLIP [Radford et al., 2021], which is trained on pairs of images and language descriptions scraped from the internet. While CLIP is trained only on images, video-language-models (VLMs) trained on videos of humans performing daily tasks such as S3D [Xie et al., 2018] or XCLIP [Ni et al., 2022] are also widely available. These models utilize language descriptions while training to supervise their visual understanding so that *semantically* similar vision inputs are embedded close together in a shared vector space. A series of recent works demonstrate that these VLMs can produce useful rewards for agent learning. Fan et al. [2022] finetune CLIP on YouTube videos of people playing Minecraft and demonstrate that the finetuned CLIP model can be used as a language-conditioned reward function to train an agent. DECKARD [Nottingham et al., 2023] then uses the fine-tuned reward function of Fan et al. [2022] to reward an agent for completing tasks proposed by a large-language model and abstract world model. PAFF [Ge et al., 2023] uses a fine-tuned CLIP model to align videos of policy rollouts with a fixed set of language skills and relabel experience with the best-aligned language label. We demonstrate that *videos* and multi-modal task specifications can be utilized to learn reward functions allowing for training agents. Additionally, we present a method to test the alignment of pretrained VLMs with deployment environments.

## 3 Method

**Overview.** RoboCLIP utilizes pretrained video-and-language models to generate rewards for online RL agents. This is done by providing a sparse reward to the agent at the end of the trajectory which describes the similarity of the agent's behavior to that of the demonstration. We utilize video-and-language models as they provide the flexibility of defining the task in terms of natural language descriptions or video demonstrations sourced either from the target robotic domain or other more naturalistic domains like human actors demonstrating the target task in their own environment. Thus, a demonstration (textual or video) and the video of an episode of robotic interaction are embedded into the semantically meaningful latent space of S3D [Xie et al., 2018], a video-and-language model pretrained on diverse videos of human actors performing everyday tasks taken from the HowTo100M

dataset [Miech et al., 2019]. The two vectors are subsequently multiplied using a scalar product generating a similarity score between the 2 vectors. This similarity score (without scaling) is returned to the agent as a reward for the episode.

**Notation.** We formulate the problem in the manner of a POMDP (Partially Observable Markov Decision Process) with $(\mathcal{O}, \mathcal{S}, \mathcal{A}, \phi, \theta, r, T, \gamma)$ representing an observation space $\mathcal{O}$, state space $\mathcal{S}$, action space $\mathcal{A}$, transition function $\phi$, emission function $\theta$, reward function $r$, time horizon $T$, and discount factor $\gamma$. An agent in state $\mathbf{s}_t$ takes an action $\mathbf{a}_t$ and consequently causes a transition in the environment through $\phi(\mathbf{s}_{t+1} \mid \mathbf{s}_t, \mathbf{a}_t)$. The agent receives the next state $\mathbf{s}_{t+1}$ and reward $r_t = r(\mathbf{o}_t, \mathbf{a}_t)$ calculated using the observation $\mathbf{o}_t$. The goal of the agent is to learn a policy $\pi$ which maximizes the expected discounted sum of rewards, i.e., $\sum_{t=0}^{T} \gamma^t r_t$. Note that all of our baselines utilize the true state for reward generation and for policy learning. To examine the effect of using a video-based reward, we also operate our policy on the state space while using the pixel observations for reward generation. Thus, $r_t$ uses $\mathbf{o}_t$ while $\pi$ uses $\mathbf{s}_t$ for RoboCLIP while for all other baselines, both $r_t$ and $\pi$ utilize $\mathbf{s}_t$. This of course is unfair to our method, but we find that in spite of the advantage provided to the baselines, RoboCLIP rewards still generate higher zero-shot success.

**Reward Generation.** During the pretraining phase, we supply the RoboCLIP reward to the agent in a sparse manner at the end of each episode. This is done by storing the video of an episode of the interaction of the agent with the environment into a buffer as seen in Figure 1. A sequence of observations of length 128 are saved in a buffer corresponding to the length of the episode. S3D is trained on videos length 32 frames and therefore the episode video is subsequently downsampled to result in a video of length $T = 32$. The video is subsequently center-cropped to result in frames of size $(250, 250)$. This is done to ensure that the episode video is preprocessed to match the specifications of the HowTo100M preprocessing used to train the S3D model. Thus the tensor of a sequence of $T$ observations $\mathbf{o}_{0:T}$ is encoded as the latent video vector $\mathbf{z}^v$ using

$$\mathbf{z}^v = S3D^{\text{video-encoder}}(\mathbf{o}_{0:T}) \tag{1}$$

The task specification is also encoded into the same space. If it is defined using natural language, the language encoder in S3D encodes a sequence of $K$ textual tokens $\mathbf{d}_{0:K}$ into the latent space using:

$$\mathbf{z}^d = S3D^{\text{text-encoder}}(\mathbf{d}_{0:K}) \tag{2}$$

If the task description is in the form of a video of length $K$, then we preprocess and encode it using the video-encoder in S3D just as in Equation (1). For intermediate timesteps, i.e., timesteps other than the final one in an episode, the reward supplied to the agent is zero. Subsequently, at the end of the episode, the similarity score between the encoded task descriptor $\mathbf{z}^d$ and the encoded video of the episode $\mathbf{z}^v$ is used as reward $r^{\text{RoboCLIP}}(T)$. Thus the reward is:

$$r^{\text{RoboCLIP}}(t) = \begin{cases} 0, & t \neq T \\ \mathbf{z}^d \cdot \mathbf{z}^v & t = T \end{cases}$$

where $\mathbf{z}^d \cdot \mathbf{z}^v$ corresponds to the scalar product between vectors $\mathbf{z}^d$ and $\mathbf{z}^v$.

**Agent Training.** Using $r^{\text{RoboCLIP}}$ defined above, we then train an agent online in the deployment environment with any standard reinforcement learning (RL) algorithm by labeling each agent experience trajectory with $r^{\text{RoboCLIP}}$ after the agent collects it. In our paper, we train with PPO [Schulman et al., 2017], an on-policy RL algorithm, however, RoboCLIP can also be applied to off-policy algorithms. After training with this reward, the agent can be zero-shot evaluated or fine-tuned on true environment reward on the target task in the deployment environment.

## 4 Experiments

We test out each of the hypotheses defined in Section 1 on simulated robotic environments. Specifically, we ask the following questions:

1. *Do existing pretrained VLMs semantically align with robotic manipulation environments?*
2. *Can we utilize natural language to generate reward functions?*
3. *Can we use videos of expert demonstrations to generate reward functions?*
4. *Can we use out-of-domain videos to generate reward functions?*

5. *Can we generate rewards using a combination of demonstration and natural language?*
6. *What aspects of our method are crucial for success?*

We arrange this section to answer each of these questions. Both RoboCLIP and baselines utilize PPO [Schulman et al., 2017] for policy learning.

**Baselines.** We use 2 state-of-the-art methods in inverse reinforcement learning: **GAIL**, or Generative Adversarial Imitation Learning [Ho and Ermon, 2016] and **AIRL** or Adversarial Inverse Reinforcement Learning [Fu et al., 2017]. Both of these methods attempt to learn reward functions from demonstrations provided to the agent. Subsequently, they train an agent using this learned reward function to imitate the expert behavior. Both methods receive a single demonstration, consistent with our approach of using a single video imitation. However, since they both operate on the ground-truth environment state, we provide them with a **trajectory of states**, instead of images, thereby providing them privileged state information that our method does not receive.

## 4.1   Domain Alignment

Pretrained vision models are often trained on a variety of human-centric activity data, such as Ego4D [Grauman et al., 2022]. Since we are interested in solving robotic tasks with view from third person perspectives, we utilize the S3D [Xie et al., 2018] VLM pretrained on HowTo100M [Miech et al., 2019], a dataset of short third-person clips of humans performing everyday activities. This dataset, however, contains no robotic manipulation data.

To analyze the alignment of the VLM to different domains, we perform a confusion matrix analysis using videos from Metaworld [Yu et al., 2020]. We collect 10 videos per task with varying values of true reward. For each video, we also collect the true reward. We then compute the RoboCLIP reward for each video using VLM alignment between the textual description of the task and the video. We visualize the correlations between the RoboCLIP and true rewards in the form of an $n \times n$ matrix where entry $(i, j)$ corresponds to the correlation between the true reward and the RoboCLIP reward generated for the $i^{\text{th}}$ task using the $j^{\text{th}}$ text description. As one can see, for a given task, the highest correlation in the matrix is for the correct textual description. We visualize one such similarity matrix in Figure 2 for Metaworld. We find that Metaworld seems to align well in the latent space of the model with a more diagonal-heavy confusion matrix. The objects are all correctly identified.

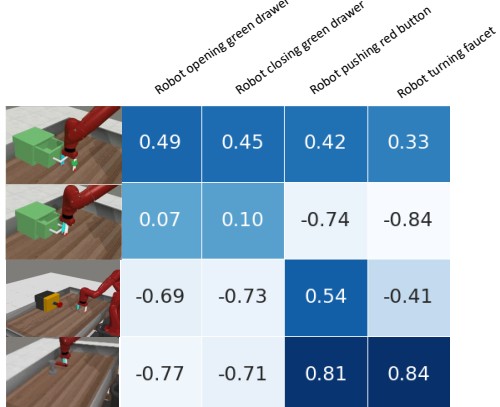

**Figure 2: Domain Alignment Confusion Matrix.** We perform a confusion matrix analysis on a subset of the data on collected on Metaworld [Yu et al., 2020] environments by comparing the pair-wise similarities between the latent vectors of the strings describing the videos and those of the videos. We find that Metaworld is well-aligned with higher scores along the diagonal than along the off-diagonal elements.

## 4.2   Language for Reward Generation

The most naturalistic way to define a task is through natural language. We do this by generating a sparse reward signal for the agent as described in Section 3: the reward for an episode is the similarity score between its encoded video and the encoded textual description of the expected behavior in the VLM's latent space. The reward is provided to the agent at the end of the episode. For RoboCLIP, GAIL, and AIRL, we first pretrain the agents online with their respective reward functions and then perform finetuning with the true task reward in the deployment environment. We perform this analysis on 3 Metaworld Environments: `Drawer-Close`, `Door-Close` and `Button-Press`. We use the textual descriptions, "*robot closing green drawer*", "*robot closing black box*", and "*robot pushing red button*" for each environment, respectively. Figure 3 plots returns on the target tasks while finetuning on the depolyment environment after pretraining (with the exception of the Dense Task Reward baseline). Our method outperforms the imitation learning baselines with online exploration in terms of true task rewards in all environments. Additionally our baselines utilize the full state information in the environment for reward generation where RoboCLIP uses only the pixels to infer state. RoboCLIP also achieves more than double zero-shot rewards in all environments — importantly,

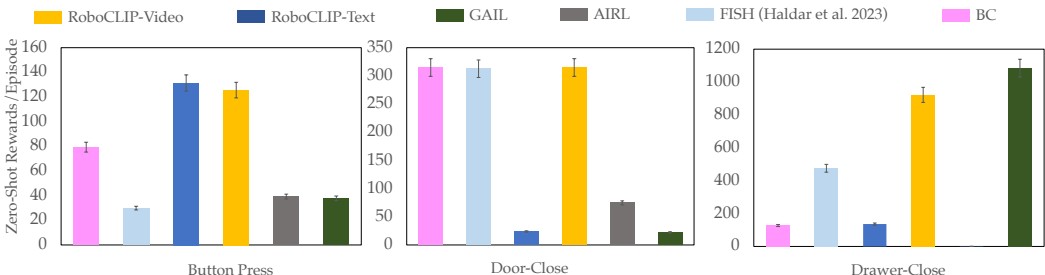

**Figure 3: Language-Conditioned Reward Generation.** The pretrained VLM is used to generate rewards via the similarity score of the encoding of an episode of interaction of an agent in its environment, $\mathbf{z}^v$ with the encoding of a task specifier $\mathbf{z}^d$ specified in natural language. We use the strings, "*robot closing black box*", "*robot closing green drawer*" and "*robot pushing red button*" for conditioning for the 3 environments respectively. We find that agents pretrained on these language-conditioned rewards outperform imitation learning baselines like GAIL [Ho and Ermon, 2016] and AIRL [Fu et al., 2017].

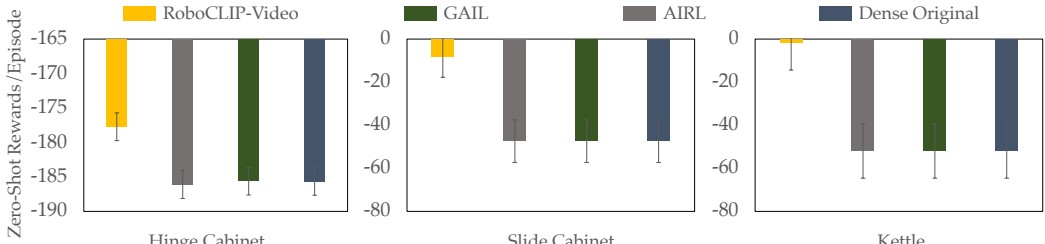

**Figure 4: Using In-Domain Videos for Reward Generation.** The pretrained VLM is used to generate rewards via the similarity score of the encoding of an episode of interaction of an agent in its environment, $\mathbf{z}^v$ with the encoding of a video demonstration of expert behavior in the same environment. The similarity score between the latent vectors is provided as reward to the agent and is used to train online RL methods. We study this setup in the `Kettle`, `Hinge` and `Slide` Tasks in the Franka Kitchen Environment [Gupta et al., 2019]. We find that policies trained on the RoboCLIP reward are able to learn to complete the task in all three setups without any need for external rewards using just a single in-domain demonstration.

the RoboCLIP-trained agent is able to complete the tasks even before finetuning on true task rewards.

## 4.3 In-Domain Videos for Reward Generation

Being able to use textual task descriptors for reward generation can only work in environments where there is domain alignment between the pretrained model and the visual appearance of the environment. Additionally, VLMs are large models often with billions of parameters making it computationally expensive to fine tune for domain alignment. The most naturalistic way to define a task in such a setting is in the form a single demonstration in the robotic environment which can be collected using teleoperation. We study how well this works in the Franka Kitchen [Gupta et al., 2019] environment. We consider access to a single demonstration per task whose video is used to generate rewards for online RL.

**Quantitative Results.** We measure the zero-shot task reward, which increases as the task object (i.e., Kettle, Slide and Hinge Cabinets) gets closer to its goal position. This reward does not depend on the position of the end-effector, making the tasks difficult. Figure 4 shows the baselines perform poorly as they generally do not interact with the target objects, while RoboCLIP is able to solve the task using the reward generated using the video of a single demonstration.

**Qualitative Results.** We find that RoboCLIP allows for mimicking the "style" of the source demonstration, with idiosyncrasies of motion from the source demonstration generally transferring to the policy generated. We find this to occur in the kitchen environment's `Slide` and `Hinge` task as seen in Figure 5. The first row of the subfigures in Figure 5 are visualizations of the demonstration video used to condition the VLM for reward generation. The bottom rows correspond to the policies that are trained with the generated rewards of RoboCLIP. As can be seen, the `Slide` demonstration consists of a wide circular arc of motion. This is mimicked in the learned policy, although the agent misses the cabinet in the first swipe and readjusts to make contact with the handle.

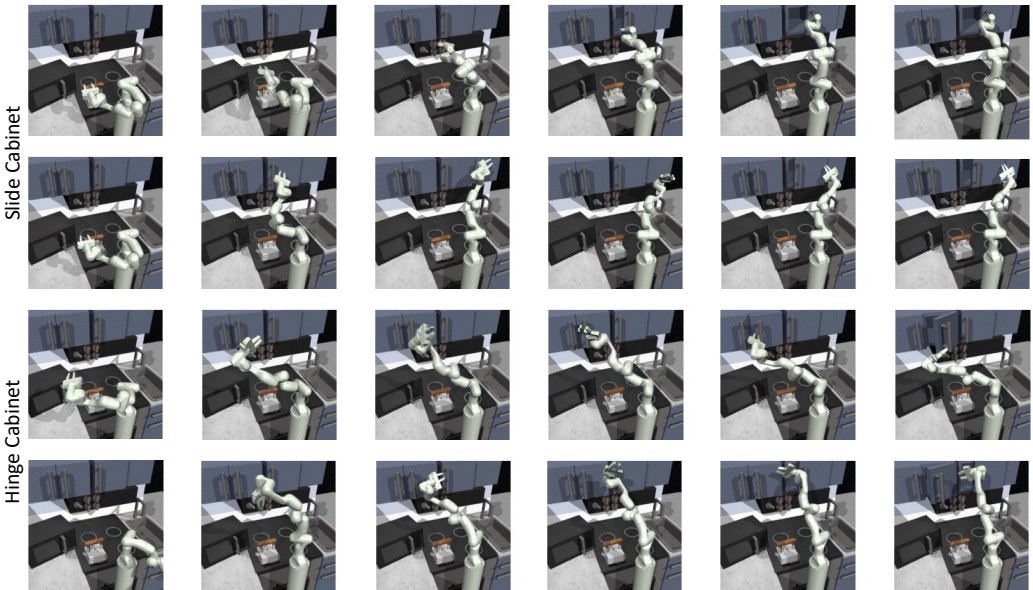

**Figure 5: Qualitative Inspection of Imitation.** The first row in each subfigure shows the visualizations of the demonstration video used for reward generation via the VLM. The second rows are videos taken from policy recovered from training on the RoboCLIP reward generated using the videos in the first rows. The quick swiping motion demonstrated in the `Slide` demonstration is mimicked well in the resultant policy while the wrist-rotational "trick-shot" behavior in the demonstration for `Hinge` appears in the resultant learned policy.

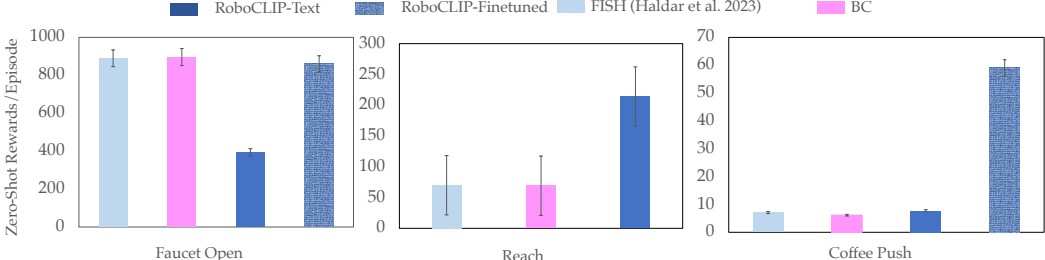

**Figure 6: Finetuning for Harder Environments:** In harder environments, like `Coffee-Push` and `Faucet-Open`, we find that RoboCLIP rewards do not solve the task completely. We test whether providing a single demonstration in the environment (using observations and actions) is enough to finetune this pretrained policy, a setup identical to our baselines. Thus, we pre-train on the RoboCLIP reward from language and then finetune using a single robotic demonstration. This improves performance by ∼ 200%. See videos on our website.

This effect is even more pronounced in the `Hinge` example where the source demonstration consists of twirling wrist-rotational behavior, which is subsequently imitated by the learned policy. The downstream policy misses the point of contact with the handle but instead uses the twirling motion to open the hinged cabinet in an unorthodox manner by pushing near the hinge. We posit that the VLMs used in RoboCLIP contain a rich latent space encoding these various motions, and so even if they cannot contain semantically meaningful latent vectors in the Franka Kitchen environments due to domain mismatch, they are still able to encode motion information allowing them to be used for RoboCLIP with a single demonstration video.

### 4.4 Out-of-Domain Videos for Reward Generation

Another natural way to define a task is to demonstrate it yourself. To this end, we try to use demonstrations of humans or animated characters acting in separate environments as task specification.

For this, we utilize animated videos of a hand pushing a red button and opening a green drawer and a real human video of opening a fridge door (see Figure 7). The animated videos are collected from stock image repositories and the human video is collected using a phone camera in our lab kitchen.

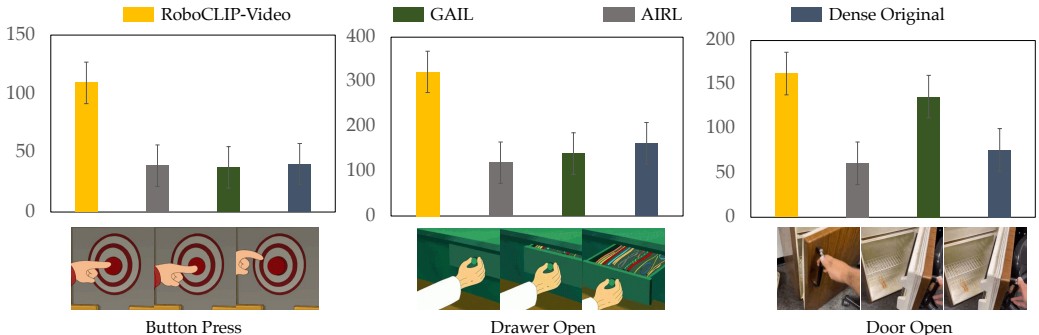

**Figure 7: Using Out-of-Domain Videos for Reward Generation.** A Pretrained Video-and-Language Model is used to generate rewards via the similarity score of the encoding of an episode of interaction of an agent in its environment, $\mathbf{z}^v$ with the encoding of a task specifier $\mathbf{z}^d$ in the form of a video of a human or an animated character demonstrating a task in their own environment. The similarity score between the latent vectors is provided as reward to the agent and is used to train online RL methods. The frames below the graphs illustrate the video used for reward generation.

Using the encodings of these video, we test out RoboCLIP in the 3 corresponding Metaworld tasks - `Button-Press`, `Drawer-Open` and `Door-Open`. We follow the same setup as in Section 4.2 by first pretraining methods with their respective reward functions and then finetuning in the deployment environment with target task reward.

We compare the performance of the policy trained with these rewards to GAIL [Ho and Ermon, 2016] and AIRL [Fu et al., 2017] trained using the same single expert demonstration as RoboCLIP on these rewards with state information. These methods are known to be data-hungry, requiring multiple demonstrations to train their reward functions. Consequently, they perform much worse than RoboCLIP, even with 2-3x worse zero-shot task performance, as can be seen from Figure 7.

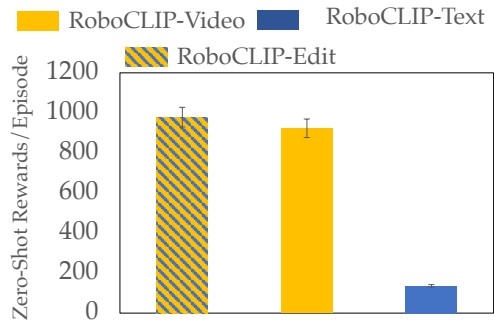

**Figure 8: Multimodal Task Specification.** We study whether video demonstrations of expert demonstrations can be used to define tasks. We use the latent embedding of a video demonstration of a robot pushing a button and subtract from it the embedding of the text "*red button*" and add to it the embedding of the text "*green drawer*". This modified latent is used to generate rewards in the Drawer-Close environment. We find that the policy trained using this modified vector outperforms string-only manipulation in the zero-shot setting.

### 4.5 Multimodal Task Specification

Using videos to specify a task description is possible when either there is access to a robot for teleoperation as in Section 4.3 or a human can demonstrate a behavior in their own environment as in Section 4.4. When these are not the case, a viable alternative is to utilize multimodal demonstrations. For example, consider a scenario where the required task is to push a drawer to close it, but only a demonstration for pushing a button is available. In this situation, being able to edit the video of the off-task demonstration is useful. This way, one can direct the agent to move its end-effectors to push the drawer instead of the button.

We do this by algebraically modifying the encoding of the video demonstration:

$$\mathbf{z}^{\text{edited}}(\text{push drawer}) = \mathbf{z}^{video}(\text{push button}) - \mathbf{z}^{text}(\text{button}) + \mathbf{z}^{text}(\text{drawer}) \tag{3}$$

where $\mathbf{z}^{\text{edited}}(\text{push drawer})$ is the vector used to generate rewards in the `Drawer-Close` environment, $\mathbf{z}^{video}(\text{push button})$ is the vector of the encoding of the video of the robot pushing a button, $\mathbf{z}^{text}(\text{button})$ is the encoding of the string *button* and $\mathbf{z}^{text}(\text{drawer})$ is the encoding of the string *drawer*. As can be seen in Figure 8, defining rewards in such a multimodal manner results in a higher zero-shot score than the dense task reward and also pretraining on the string-only task reward.

### 4.6 Finetuning

In harder environments, and with rewards from OOD videos and language, the robot policy sometimes approaches the target object, but fails to complete the task. Thus, we tested whether providing a single demonstration (using observations and actions) was enough to finetune this pretrained policy.

Thus, for this experiment we first (1) pretrain on the RoboCLIP reward from human videos or language descriptions and then (2) finetune using a single demonstration. As seen in Figure 6, we find that this converts each of the partially successful policies into complete success and improves the rewards attained by the policies by 200%. This fine-tuning setup is especially useful in harder tasks like like `Coffee-Push` and `Faucet-Open` and is competitive with state-of-the-art approaches like FISH [Haldar et al., 2023].

### 4.7 Ablations

Finally, we investigate the effects of various design decisions in RoboCLIP. First, we study the effect of additional video demonstrations on agent performance. We also examine the necessity of using a pre-trained VLM. Recent works like RE3 [Seo et al., 2021, Ulyanov et al., 2018] have shown that randomly initialized networks often contain useful image priors and can be used to supply rewards to agents to encourage exploration. Therefore, we test whether a *randomly initialized* S3D VLM can supply useful pretraining rewards in the in-domain video demonstration setup as in Section 4.3. Finally, we study our choice of pre-trained VLM. We examine whether a pretrained CLIP [Radford et al., 2021], which encodes single images instead of videos and was trained on a different dataset from S3D, can be used to generate rewards for task completion. In this setup, we record the last image in an episode of interaction of the agent in its environment and feed it to CLIP trained on ImageNet [Russakovsky et al., 2015] (i.e., not trained on videos). We then specify the task in natural language and use the similarity between the embeddings of the textual description of the task and the final image in the episode to generate a reward that is fed to the agent for online RL.

As seen in Figure 9, using a single video demonstration provides the best signal for pre-training. We posit that our method performs worse when conditioned on multiple demonstrations as the linear blending of multiple video embeddings, which is used due to the scalar product, does not necessarily correspond to the embedding of a successful trajectory.

Crucially, we also find that using the static image version of CLIP does not provide any useful signal for pretraining. The zero-shot performance is very poor, which we posit is because it does not contain any information about the dynamics of motion and task completion although it contains semantic meaning about objects in the frame. On the other hand, video contrastive learning approaches do contain this information. This is further evidenced by the fact that inspite of poor domain alignment between Franka Kitchen and the VLM, we find that encodings of in-domain video demonstrations are still good for providing a pretraining reward signal to the agent.

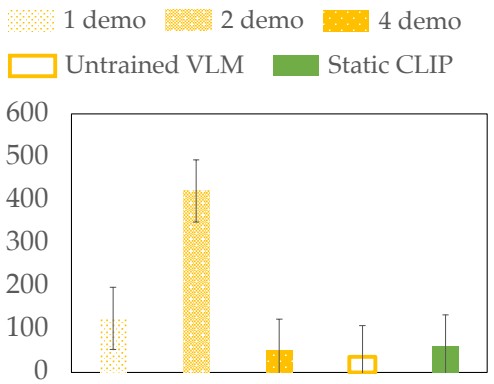

**Figure 9: Ablations.** We study the effects of varying the number of demonstrations provided to the agent can have on downstream rewards. We also study the effects of the training provided to the VLM on the downstream rewards. Finally, we study whether using CLIP trained on static images provides good rewards for pretraining.

## 5 Conclusion

**Summary.** We studied how to distill knowledge contained in large pretrained Video-and-Language-Models into online RL agents by using them to generate rewards. We showed that our method, RoboCLIP, can train robot policies using a single video demonstration or textual description of the task, depending on how well the domain aligns with the VLM. We further investigated alternative ways to use RoboCLIP, such as using out-of-domain videos or multimodal demonstrations. Our results showed RoboCLIP outperforms the baselines in various robotic environments.

**Limitations and Broader Impact.** Since we are using VLMs, the implicit biases within these large models could percolate into RL agents. Addressing such challenges is necessary, especially since it is

unclear what the form of biases in RL agents might look like. Currently, our method also faces the challenge of stable finetuning. We find that in some situations, finetuning on downstream task reward results in instabilities as seen in the language conditioned reward curve in Figure 8. This instability is potentially due to the scale of rewards provided to the agent. Rewards from the VLM are fairly low in absolute value and subsequently, the normalized Q-values in PPO policies are out-of-shape when finetuned on task rewards. In our experiments, this is not a big problem since the RoboCLIP reward is already sufficient to produce policies that complete tasks without any deployment environment finetuning, but this will be essential to solve when deploying this for longer horizon tasks.

Another limitation of our work is that there is no fixed length of pretraining. Our current method involves pretraining for a fixed number of steps and then picking the best model according to the true task reward. This is of course difficult when deploying RoboCLIP in a real-world setup as a true reward function is unavailable and a human must monitor the progress of the agent. We leave this for future work.

## Acknowledgments and Disclosure of Funding

This work was supported by DARPA (HR00112190134), C-BRIC (one of six centers in JUMP, a Semiconductor Research Corporation (SRC) program sponsored by DARPA), the Army Research Office (W911NF2020053) and the Office of Naval Research (ONR) (#N00014-21-1-2298 and #N00014-21-1-2685). The authors affirm that the views expressed herein are solely their own, and do not represent the views of the United States government or any agency thereof. Thanks to Karol Hausman, Sergey Levine, Ofir Nachum, Kuang-Huei Lee and many others at Google DeepMind for helpful discussions.

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
