# OpenReview forum: "RoboCLIP: One Demonstration is Enough to Learn Robot Policies"
_NeurIPS.cc/2023/Conference — NeurIPS 2023 poster_

### Official Review · Reviewer_EmDU · 2023-07-02

**Soundness:** 2 fair
**Presentation:** 2 fair
**Contribution:** 2 fair
**Rating:** 5
**Confidence:** 4

**Summary:**

This study seeks to address the problem of data efficiency in imitation learning, aiming to train a robot to perform manipulation tasks after only one demonstration. In order to accomplish this, the research proposes using a pre-trained visual language model (VLM) to encode the demonstration video or task description text from the expert. The use of the dot product between the learnt and demonstrated embeddings would then serve as a reward function. Experiments are conducted in Metaworld and the Franka simulation, with tasks including "closing a door," "closing a drawer," and "pressing a button."

**Strengths:**

- The idea of utilizing a pretrained visual language model to encode expert demonstrations or text descriptions as a reward function is neat and straightforward.

- This work conducts an extensive amount of robot experiments in simulation including"closing a door", "pressing a button", "Slide Cabinet " and so on.

**Weaknesses:**

- The use of dot product between expert embeddings and learned policy embeddings as the objective reward function is not adequately explained. The rationale behind this decision and its effectiveness would benefit from further elaboration.

- The Domain Alignment Confusion Matrix reveals a poor similarity alignment between the text description and the video from the robot camera. This implies a potential issue, where goal descriptions in text form may not correspond accurately to the video data.


- Moreover, the experimental performance demonstrated in the accompanying video is weak, with numerous instances of failure.


- The selected baseline is outdated. GAIL was introduced in 2016 and AIRL was in 2017.

- The proposed method fails to improve and instead performs significantly worse when presented with an increased number of expert demonstrations.

**Questions:**

Q1: In Figure 4, what is the reason behind GAIL, AIRL, and a method called "Task Reward" reflecting an identical reward across all three tasks?

**Limitations:**

See Weaknesses and Questions.

---

> ### Author Rebuttal · Authors · 2023-08-09
>
> Thanks for your detailed review! Addressing your comments has improved the paper significantly! :)
> We address your concerns below:
> 1. **“Why use dot products between embeddings as the reward function?”**: We will add the following to the text: VLMs such as VideoCLIP and S3D are pretrained using a cosine similarity loss that aligns encodings of different modalities such as language and videos. This is done by encoding both modalities into latent spaces of the same dimensions and calculating a dot product between the vectors of each modality. The resultant dot product (a scalar) measures the similarity between the embeddings, and the model is trained to increase similarity between paired videos and language labels.
> Therefore, this dot product naturally serves as a reward function for an agent: we compute the dot product between embeddings of agent videos and the given task specification.
> 2. **“Outdated Baselines?”**: The reviewer is right that there has been considerable work on learning from demonstrations since the publication of our baselines. We have therefore added 2 more baselines (FISH [1], **published ~15 days ago at RSS 2023**, and Behavior Cloning), the most effective of which is FISH. FISH demonstrates how it is possible to learn robot policies using 1 min of demo data - thereby also operating in the same low data regime. However, we find that in our extremely low data regime of **one demonstration**, RoboCLIP outperforms even FISH by 3x in 2 out of 3 environments, and matches its performance in 1 (see Fig 10 in the rebuttal doc). This comparison highlights RoboCLIP’s performance, even compared to the latest SOTA learning from demonstration baselines. Summary:
> \begin{array} {|r|r|} \text{Task} & \text{RoboCLIP-V} & \text{RoboCLIP-T} & \text{FISH} & \text{BC} \\\  \text{Door Close} & \textbf{314.8915735} & 24.08013125 & 312.7388271 & 314.8853879 \\\  \text{Drawer Close} & \textbf{922.5} & 136.25 & 475.2357946 & 127.44404 \\\  \text{Button Press} & 125.6114189 & \textbf{131.365798} & 29.8517859 & 79.35955141 \\   \end{array}
> 3. **“Numerous instances of failure in the accompanying video?”**: We agree, thanks for pointing this out! Indeed, especially with rewards from OOD videos and language, the robot policy often approaches the target object, but fails to complete the task. Thus we now added an **additional experiment**: We tested whether providing a single demonstration, the same demonstration used to train the baselines (using observations and actions), was enough to finetune this pretrained policy. Thus, for this experiment we first (1) pretrain on the RoboCLIP reward from human videos or language descriptions and then (2) finetune using a single demonstration.
> Through this experiment, **RoboCLIP policies successfully complete the tasks across all 7 Metaworld setups.** In the new challenging environments, it results in ~3x higher rewards in 2 out of 3 environments and matching the performance on 1. **This setup solves all of the failure policies seen previously.** Please see additional videos of this experiment on the website (“Finetuning with One Demo” Section) and Fig 10 in the rebuttal document. We find that this setup provides overwhelming evidence that, **with RoboCLIP, one demonstration is enough to learn a policy**. Summary of new environments:
> \begin{array} {|r|r|} \text{Task} & \text{RoboCLIP-Finetuned} & \text{FISH} & \text{BC} \\\  \text{Faucet Open} & 859.9849646 & 889.4014539 & 896.2618905 \\\  \text{Coffee Push} & \textbf{59.00943929} & 7.099759383 & 6.183762854 \\\  \text{Reach} & \textbf{214.0404179} & 70.05093234 & 69.42764688 \\   \end{array}
>
> 4. **"RoboCLIP performs worse with additional expert demos? (Sec 4.6)?":**
> We added an **additional experiment** using 2 demonstrations, finding that it actually outperforms using either a single demonstration or 4 demonstrations. However, using 4 demonstrations does indeed reduce performance. We believe that this is because the convex combination of multiple demonstrations may not result in a successful agent trajectory, thus this could be why using 4 demonstrations results in worse performance. We will add this conjecture and experiment to Section 4.6. Summary:
> \begin{array} {|r|r|} \text{RoboCLIP-V 1 Demo} & \text{RoboCLIP-V 2 Demos} & \text{RoboCLIP-V 4 Demos} \\\  125.6114189 & \textbf{421.5395214} & 51.34670371 \\   \end{array}
> 5. **“Domain alignment confusion matrix shows poor similarity between certain text descriptions and videos?”**: The alignment metric for the confusion matrix was calculated by aligning a single demonstration video’s embedding with that of a language prompt. This is a noisy estimate due to the small sample size and this metric does not align with task progress.
> Therefore, we calculate a **new confusion matrix** by calculating the correlation between true and RoboCLIP reward over 10 agent trajectories. We found that this metric actually demonstrates alignment: the resultant matrix is diagonal-heavy (see below). This demonstrates that for each of the metaworld environments, the RoboCLIP reward does indeed correlate with the true reward.
> \begin{array} {|r|r|} \text{Video || Text} & \text{Robot Opening Green Drawer} & \text{Robot Closing Green Drawer} & \text{Robot Pushing Red Button} & \text{Robot Turning Handle}\\\  \text{Open drawer videos} & \textbf{0.49} & 0.45 & 0.42 & 0.33 \\\  \text{Drawer Close videos} & 0.07 & \textbf{0.1} & -0.74 & -0.84 \\\  \text{Button Press Videos} & -0.6925 & -0.7309	 & \textbf{0.5351} & -0.41 \\\  \text{Faucet Videos} & 	-0.77 & -0.71	 & 0.81 & \textbf{0.84} \\   \end{array}
>
>
> [1] Haldar, Siddhant, et al. "Teach a Robot to FISH: Versatile Imitation from One Minute of Demonstrations." arXiv preprint arXiv:2303.01497 (2023)

---

> > ### Comment · Reviewer_EmDU · 2023-08-17
> > **Reply to Authors**
> >
> > Thanks for your rebuttal. I am still not fully convinced. The learned robot's performance is not satisfactory.
> >
> > For instance, in the task entitled "Robot closing green drawer", the video shows the robot moving away from the drawer, while the drawer magically closes on its own. This demonstrates an improper execution of the task. In addition,  "Out-of-Domain Videos to Generate Rewards" seems to fail on multiple tasks.
> >
> > The new confusion matrix is still problematic, as the robot is still confused by the text "close drawer" and the video "open drawer".

---

> > > ### Author Response · Authors · 2023-08-17
> > > **Reply**
> > >
> > > Thanks for getting back to us!
> > > 1. Sorry but we are not sure you have seen the updated videos section. We have therefore relabeled the correct section to look at as **Finetuning with One Demonstration (for Reviewer EmDU)**. This is also now the first section on the website after the title. We would have loved to provide a direct link but the rules don't let us :(. Crucially, the "Out-of-Domain Videos to Generate Rewards" section corresponds to using **zero robotic demonstrations**. Therefore failures are expected (the main claim of the paper is that 1 robotic demo is needed, not zero). When provided with **one robotic demonstration, all of the failures are resolved**. In this correct section (please search for **EmDU** on the site and this will take you to the section to view), you will find the failure policy videos on the left column which correspond to the policies learned with **zero** robot demos (the same ones learned from the OOD videos). On the right are the same policies fine-tuned using **1 demo**. As you can see, with one demo, all policies complete the task. This validates the main claim of the paper, **that in all experimental setups, one demo is enough to learn a policy using RoboCLIP**.
> > > 2. The reason why the drawer close video looks the way it does is because of rendering. The robot makes contact with the drawer in literally the first few steps of the rollout. Since closing the drawer requires very little force, this interaction is enough to finish the task. In fact, the video uploaded on the site **receives a true reward of 1050** for this seed due to how efficiently it closes the drawer (**baselines get about ~300** and the scripted policy defined in Metaworld gets 481.34). We posit that the reason the robot moves away from the drawer is that the VLM has never seen drawer-opening videos with a robot in them. Thus to maximize the RoboCLIP reward and align the demo to the training data it has seen, it moves away from the drawer ASAP once the required force to close the drawer has been applied.
> > > 3. The confusion matrix **is diagonal heavy**, i.e., for each of the tasks, the correct string has the highest correlation with the correct videos. The way to look at the matrix is row-wise. This is because we want to ensure that for a given task (i.e., row corresponding to the videos), the highest entry should correspond to the string describing the task. Thus from the data, there is **no confusion**. We posit that there is some correlation between the remaining pairs because the VLM attends to the "green drawer" in the text and the green drawer object in the video resulting in a correlation. But for each task, **the highest correlation entry corresponds to the correct pair.** Further, the main reason to believe the method works is the performant policies obtained across multiple repeated seeds.

---

### Official Review · Reviewer_S3ve · 2023-07-06

**Soundness:** 3 good
**Presentation:** 3 good
**Contribution:** 3 good
**Rating:** 6
**Confidence:** 3

**Summary:**

RoboCLIP is an innovative online imitation learning method that addresses the challenging problem of reward specification in reinforcement learning. Unlike traditional approaches that rely on expert supervision and extensive data, RoboCLIP utilizes a single demonstration, either in the form of a video or a textual description, to generate rewards without the need for manual reward function design. It can even leverage out-of-domain demonstrations, eliminating the requirement for demonstrations from the same domain as deployment.

Basically the method compares the embedding between the video generated by the policy and video given to it for imitation to get a reward for training an RL agent. The idea is simple and intuitive.

**Strengths:**

Although the idea is simple, the authors verify through experimentation that learning policies from videos is possible.
The results are great and contribute to advance in robot policy learning from videos.

**Weaknesses:**

Out-of-distribution results are underwhelming. Maybe a future direction could to focus more on the object state to train better policies.

No real robot results :(
The whole point of something like this should be easy applicability to the real world.

**Questions:**

I don't have any questions for the authors.

**Limitations:**

limitations are discussed adequately.

---

> ### Author Rebuttal · Authors · 2023-08-09
>
> Thanks for your comments! We're glad that you would like to see our work published!
> We address your concerns below:
> 1. **“Out-of-distribution results are underwhelming? (Sec 4.4)”**: Indeed, especially with rewards from OOD videos and language, the robot policy often approaches the target object, but fails to complete the task. Thus we now added an **additional experiment**: We tested whether providing a single demonstration, the same demonstration used to train the baselines (using observations and actions), was enough to finetune this pretrained policy. Thus, for this experiment we first (1) pretrain on the RoboCLIP reward from human videos or language descriptions and then (2) finetune using a single demonstration. Through this experiment, RoboCLIP policies successfully complete the tasks **across all setups** resulting in ~3x higher rewards in 2 out of 3 environments and matching the performance on 1.  **This setup solves all of the failure policies seen previously.** Please see additional videos of this experiment on the website (“Finetuning with One Demo”) and Fig 10 in the rebuttal document. We find that this setup results in overwhelming evidence that, with RoboCLIP, one demonstration is enough to learn a policy. Summary:
> \begin{array} {|r|r|} \text{Task} & \text{RoboCLIP-Finetuned} & \text{FISH} & \text{BC} \\\  \text{Faucet Open} & 859.9849646 & 889.4014539 & 896.2618905 \\\  \text{Coffee Push} & \textbf{59.00943929} & 7.099759383 & 6.183762854 \\\  \text{Reach} & \textbf{214.0404179} & 70.05093234 & 69.42764688 \\   \end{array}
> 2. **“No real robot results?"**:We agree that real robot results would be exciting. We avoided training on real robots for safety reasons due to unstable behaviors that could arise when pre-training the policy on a real robot with VLM rewards. However the aforementioned single demonstration fine-tuning experiment highlights the possibility of: first pre-training with RoboCLIP in simulation, then finetuning on the real robot using **robot demonstrations**. We leave investigating this to follow-up work.

---

### Official Review · Reviewer_Dcaj · 2023-07-09

**Soundness:** 2 fair
**Presentation:** 3 good
**Contribution:** 2 fair
**Rating:** 6
**Confidence:** 4

**Summary:**

This paper presents a novel approach to leverage pretrained Vision-Language Models (VLMs) for training Reinforcement Learning (RL) agents by utilizing them to generate rewards. Specifically, the authors employ S3D, pretrained on HowTo100M, as the backbone and measure the dissimilarity between online agent experience and task demonstrations to serve as the reward signal for training task-specific RL policies. They demonstrate the effectiveness of their proposed method on the Metaworld Environment and the Franka Kitchen Environment.

**Strengths:**

1. This paper addresses an important and widely recognized problem, which is the grounding of VLMs.
2. The idea behind the proposed method is straightforward and intuitive.
3. The author thoroughly examines the performance of the proposed approach from six different perspectives: alignment, text-to-reward mapping, video-to-reward mapping, generalization to out-of-domain videos, decompositional generalizability, and an ablation study.

**Weaknesses:**

I have some major concerns about the experiment section of this paper. There are several results that are not convincing and can be quite confusing.

1. The domain alignment between the demonstration and text description, as mentioned by the author, is indeed unsatisfactory. For instance, the policy should learn to close the green drawer instead of opening it when the task descriptor is "open green drawer." Using this misleading reward can lead to misleading learning outcomes.

2. In the language-to-reward section (Section 4.2), the author only demonstrates the performance on three "easy" tasks, which is not sufficiently convincing. Additionally, it is unclear why the proposed method outperforms dense task rewards. The dense task reward can be considered an oracle, so it raises questions about whether the performance improvement is due to insufficient training epochs or simply because the rewards themselves are inadequate.

3. The qualitative results presented in Section 4.3 are challenging to comprehend. It is difficult to understand how the robot can open the cabinet without even touching it. While I can imagine what happened in between, the visualizations provided in the paper do not offer enough information to support a clear understanding of the process.

4. Regarding multimodal task specification, it is puzzling why RoboCLIP-video (with a different object) yields the best performance instead of RoboCLIP-Edit. One possible interpretation is that the robot is primarily learning push-like behaviors from a video demonstration. However, this raises concerns about the overall effectiveness and usefulness of the edit method in achieving the intended goals.

**Questions:**

1. What is RoboCLIP-Video in Section 4.2?
2. In Section 4.6, how did you use 4 demos to generate one reward for training?

**Limitations:**

I appreciate that the author acknowledges the limitation of unstable fine-tuning in their work.

---

> ### Author Rebuttal · Authors · 2023-08-09
>
> Thank you very much for your insightful comments! Addressing them has made this work much better! :)
> We address each of your concerns below:
> 1. **“VLM confuses open and close, the domain alignment is not good?”**: The alignment metric for the confusion matrix was calculated by aligning a single demonstration video’s embedding with that of a language prompt. This is a noisy estimate due to the small sample size and this metric does not account for task progress.
> Therefore, we calculate a **new confusion matrix** by calculating the correlation between true and RoboCLIP reward over 10 agent trajectories. We found that this metric actually demonstrates alignment: the resultant matrix is diagonal-heavy (see below). This demonstrates that for each of the metaworld environments, the RoboCLIP reward does indeed correlate with the true reward.
> \begin{array} {|r|r|} \text{Video \\\ Text} & \text{Robot Opening Green Drawer} & \text{Robot Closing Green Drawer} & \text{Robot Pushing Red Button} & \text{Robot Turning Handle}\\\  \text{Open drawer videos} & \textbf{0.49} & 0.45 & 0.42 & 0.33 \\\  \text{Drawer Close videos} & 0.07 & \textbf{0.1} & -0.74 & -0.84 \\\  \text{Button Press Videos} & -0.6925 & -0.7309	 & \textbf{0.5351} & -0.41 \\\  \text{Faucet Videos} & 	-0.77 & -0.71	 & 0.81 & \textbf{0.84} \\   \end{array}
> 2. **“Not enough tasks evaluated in Sec 4.2?”** : Thanks for your suggestion! We have **added 3 visually distinct environments** in metaworld as additional experimental domains for this rebuttal  - (1) Coffee-push which involves the picking and placing of a very small coffee mug. The coffee mug is about half the size of the robot end-effectors – manipulating this object requires significant precision (please see representative videos on the website: “Finetuning with One Demonstration Section”). The other environments are (2) Faucet, which requires the robot to turn a thin handle and (3) Reach, where the agent must reach a tiny dot in space. (1) and (2) are significantly more challenging than the existing environments.
> We also run 2 more baselines (FISH [1] and Behavior Cloning) on these new environments. In more complex environments like (1), FISH fails to learn successful policies while RoboCLIP performs well–outperforming FISH by 3x. Summary of results:
> \begin{array} {|r|r|} \text{Task} & \text{RoboCLIP-Finetuned} & \text{FISH} & \text{BC} \\\  \text{Faucet Open} & 859.9849646 & 889.4014539 & 896.2618905 \\\  \text{Coffee Push} & \textbf{59.00943929} & 7.099759383 & 6.183762854 \\\  \text{Reach} & \textbf{214.0404179} & 70.05093234 & 69.42764688 \\   \end{array}
> 3. **“Why does the method outperform the dense task reward baseline? (Sec 4.2)”**: To clarify, the dense task reward corresponds to the reward produced by an untrained agent. This baseline allows us to compare the performance to an agent that trains on the task for the same amount of time without any prior training. This baseline indicates that the task is not trivial and that the pretraining is crucial. Thanks for pointing out this confusion, we will clarify the description in Section 4.
> 4. **“Qualitative results in Sec 4.3 hard to understand? (Fig 5)”**: To clarify the qualitative results in Section 4.3, here is what we will add to the text: The first row of each of the subfigures is a  visualization of the demonstration video used to condition the VLM for reward generation. The bottom rows correspond to the policies that are obtained by training with the resultant RoboCLIP reward obtained from the above demonstration. As can be seen, the Slide demonstration consists of a wide circular arc of motion. This is mimicked in the learned policy, although the agent misses the cabinet in the first swipe and readjusts to make contact with the handle. This effect is even more pronounced in the Hinge example where the source demonstration consists of twirling wrist-rotational behavior, which is subsequently imitated by the learned policy. The downstream policy misses the point of contact with the handle but instead uses the twirling motion to open the hinged cabinet in an unorthodox manner by pushing near the hinge. We also have gifs demonstrating this exact visualization on our website under the “Style Transfer” section. The link to the site can be found in the abstract.
> 5. **“Why does RoboCLIP-Video, using a different object, perform better than RoboCLIP-Edit?”** To clarify: the RoboCLIP video datapoint here is with the target object, in this case, the drawer and this datapoint is meant to serve as a “skyline”. The point of this experiment is to demonstrate that editing can also allow one to reach the performance obtained when using a “true” demonstration. We will make this clearer in Section 4.5.
>
> [1] Haldar, Siddhant, et al. "Teach a Robot to FISH: Versatile Imitation from One Minute of Demonstrations." arXiv preprint arXiv:2303.01497 (2023).
>
>
> **QUESTIONS:**
> 1. **“What is RoboCLIP-Video?”**
> RoboCLIP video in section 4.2 corresponds to the policy obtained by training on the video reward generated using an in-domain video demonstration within metaworld. We will clarify this in the text in Section 4.2.
> 2. **“Sec 4.6: How did we use 4 demos to generate 1 reward for training?”**
> We used 4 demonstrations to generate the reward by encoding each of the demonstrations into the latent space. Then we calculated the similarity between the video of an episode and each of the 4 demonstrations and calculate the reward as a sum of each of the 4 similarity scores. We will update Sec 4.6 to explicitly state the above.

---

> > ### Comment · Reviewer_Dcaj · 2023-08-20
> > **Official Comment by Reviewer Dcaj**
> >
> > I would thank the authors for their thoughtful rebuttal. The points raised in the rebuttal have effectively addressed the majority of my concerns. Notably, the revised confusion matrix reduced the noise and showed a more pronounced diagonal distribution. However, I must emphasize that this work maintains an overly direct approach and exists within a landscape containing numerous similar works, which could limit its potential for significant impact. As a result, I'd love to raise my evaluation score to weak accept.

---

### Official Review · Reviewer_QrrK · 2023-07-10

**Soundness:** 3 good
**Presentation:** 4 excellent
**Contribution:** 3 good
**Rating:** 6
**Confidence:** 3

**Summary:**

This paper proposes using the similarity between two embedded spaces, one from real interaction and the other from visual/textual interaction, to develop the rewards.

**Strengths:**

It's a well-written paper. The idea is simple and clearly stated. The proposed method has been sufficiently evaluated on realistic simulation settings.

**Weaknesses:**

It is possible to make the paper more relevant to machine learning as it looks like a robotics paper. The intuition/rationale behind the simple sparse reward structure is not properly explained.

**Questions:**

1) Can authors comment why the paper is more suitable to the NeurIPS readership than CoRL, RSS, ICRA, etc. readership?
2) Can the authors comment why the simple sparse reward of the two embeddings would work? Given that the video demonstrations are not anywhere closer to the real robot interaction, this reviewer is having a hard time understanding how the model works.
3) Looking at Fig 2, it looks like the model is not great at distinguishing "opening" from "closing." Why has the model not learned that concept?
4) What is the cause of variability of rewards in Fig 3? Is it mainly due to how close the video to the real or some other reason? Since the video embedding seems to be significantly more useful, what does text serve? Are the authors trying to convey that even using text (which is easier) is fine if we don't have a video.

**Limitations:**

Adequate

---

> ### Author Rebuttal · Authors · 2023-08-09
>
> Thank you for your thoughtful review! We are glad you would like to see our work published! :)
> We address your questions below:
> 1. **“Why submit to NeurIPS over robot conferences?”**:  Thanks for your suggestion! Sequential decision-making is a generally useful problem to the community and recent advances in machine learning borrow heavily from robotics literature. Consider, for instance, reinforcement learning from human feedback (RLHF), a method that is utilized to finetune most of today’s large language models (LLMs). RLHF [1] when finetuning LLMs utilizes PPO [2], both of which were primarily robotics works now applicable to LLMs. Similarly, we believe that our work could inspire related fields to utilize our approach (e.g., the policy in our work could be replaced by an LLM and the pre-trained VLM could be a hate speech detector and the LLM could be finetuned to be prevented from generating hate speech). Furthermore, robotics works like ours [4, 5] are commonly published at ICML, ICLR, NeurIPS, and related conferences.
>
> 2. **“Why does the sparse VLM embedding reward work?"**: The VLM embedding reward encourages the agent to produce trajectories that align semantically with the given demonstration. The VLM is able to do this due to its pre-training objective (embedding videos with similar captions close to each other). Due to this semantic alignment, it will encode successful agent trajectories of say, opening a door, close to the video demo of opening a door or to textual description *opening door*. This distance in the embedding space very naturally translates as a reward.
>
> 3. **“Fig 2: VLM does not distinguish opening and closing in confusion matrix?”**: The alignment metric for the confusion matrix in the current version was calculated by aligning a single demonstration video’s embedding with that of a language prompt. Thinking about the reviewer comments, we noted how this is a noisy estimate of alignment due to the small sample size (of 1) and that this metric does not align with task progress.
> Therefore, we calculate a **new confusion matrix** by calculating the correlation between true and RoboCLIP reward over 10 agent trajectories. We found that this metric actually demonstrates alignment: the resultant matrix is diagonal-heavy (see below). This demonstrates that for each of the metaworld environments, the RoboCLIP reward does indeed correlate with the true reward.
> \begin{array} {|r|r|} \text{Video || Text} & \text{Robot Opening Green Drawer} & \text{Robot Closing Green Drawer} & \text{Robot Pushing Red Button} & \text{Robot Turning Handle}\\\  \text{Open drawer videos} & \textbf{0.49} & 0.45 & 0.42 & 0.33 \\\  \text{Drawer Close videos} & 0.07 & \textbf{0.1} & -0.74 & -0.84 \\\  \text{Button Press Videos} & -0.6925 & -0.7309	 & \textbf{0.5351} & -0.41 \\\  \text{Faucet Videos} & 	-0.77 & -0.71	 & 0.81 & \textbf{0.84} \\   \end{array}
>
> 4. **“Is the variance of rewards between Fig3 due to domain differences?”**:Yes, as the reviewer has predicted, domain alignment is the main cause! The video-conditioned reward is obtained from the same modality as the environment interactions, therefore resulting in higher rewards.
>
> 5. **“Why use text embeddings if video embeddings work better?”**:  Yes! The point of that experiment is to show that if a video demonstration cannot be collected, using a text description of the task can still enable agent learning. This is crucial in applications where in-domain demonstrations are unavailable - for instance, teaching a robot to perform tasks in a location inaccessible to humans.
>
> [1] Deep Reinforcement Learning from Human Preference. Christiano et al. 2017.
>
> [2] Proximal Policy Optimization Algorithms Schulman et al. 2017.
>
> [3] GPT-4 Technical Report
>
> [4] Dynamics-Aware Unsupervised Discovery of Skills, Sharma et al, ICLR 2020.
>
> [5] Data-Efficient Hierarchical Reinforcement Learning, Nachum et al, NeurIPS 2018.

---

### Author Rebuttal · Authors · 2023-08-09

We would like to thank the reviewers for their insightful comments and suggestions. They have helped improve the paper significantly and we are grateful for their time and effort :). We have run 4 additional experiments to address their concerns. We summarize the findings below. For details, please see the rebuttal doc.
1. **3 New Environments**: We have added 3 new environments: (1) Coffee-push which involves the picking and placing of a very small coffee mug. The coffee mug is about half the size of the robot end-effectors – manipulating this object requires significant precision (please see representative videos on the website: “Finetuning with One Demonstration Section”). The other environments are (2) Faucet, which requires the robot to turn a thin handle and (3) Reach, where the agent must reach a tiny dot in space. (1) and (2) are significantly more challenging than the existing environments.
2. **Additional Baselines:** We have added 2 more baselines (FISH [1], published **~15 days ago** at RSS 2023, and Behavior Cloning), the most effective of which is FISH. FISH demonstrates how it is possible to learn robot policies using 1 min of demo data - thereby also operating in the same low data regime. However, we find that in our extremely low data regime of **one demonstration**, RoboCLIP outperforms even FISH by 3x in 2 out of 3 environments, and matches its performance in 1 (see Fig 10 in the rebuttal doc). An abridged version of the results are below:
\begin{array} {|r|r|} \text{Task} & \text{RoboCLIP-V} & \text{RoboCLIP-T} & \text{FISH} & \text{BC} \\\  \text{Door Close} & \textbf{314.8915735} & 24.08013125 & 312.7388271 & 314.8853879 \\\  \text{Drawer Close} & \textbf{922.5} & 136.25 & 475.2357946 & 127.44404 \\\  \text{Button Press} & 125.6114189 & \textbf{131.365798} & 29.8517859 & 79.35955141 \\   \end{array}
3.  **Additional Finetuning Experiment** With rewards from OOD videos and language, the robot policy often approaches the target object, but fails to complete the task. Thus we now added an additional experiment: We tested whether providing a single demonstration, the same demonstration used to train the baselines (using observations and actions), was enough to finetune this pretrained policy. Thus, for this experiment we first (1) pretrain on the RoboCLIP reward from human videos or language descriptions and then (2) finetune using a single demonstration.
Through this experiment, RoboCLIP policies **successfully complete the tasks across all 7 Metaworld setups converting all previous failure policies into successes**. In the new challenging environments, it results in ~3x higher rewards in 2 out of 3 environments and matching the performance on 1. See additional videos of this experiment on the website (“Finetuning with One Demo”) and Fig 10 in the rebuttal document. We find that this setup provides overwhelming evidence that, **with RoboCLIP, one demonstration is enough to learn a policy.** Summary below:
\begin{array} {|r|r|} \text{Task} & \text{RoboCLIP-Finetuned} & \text{FISH} & \text{BC} \\\  \text{Faucet Open} & 859.9849646 & 889.4014539 & 896.2618905 \\\  \text{Coffee Push} & \textbf{59.00943929} & 7.099759383 & 6.183762854 \\\  \text{Reach} & \textbf{214.0404179} & 70.05093234 & 69.42764688 \\   \end{array}
4. **Improved Domain Alignment Confusion Matrix Calculation:** In the current version of the paper, the domain alignment matrix was calculated by encoding a video of a demonstration of a target task and its language description and calculating the similarity score between them using a pre-trained VLM. This metric: (1) was noisy: due to a small sample size of 1, (2) Did not capture a notion of task completion. Thus, we calculated a new domain alignment matrix by calculating the correlation between true and RoboCLIP reward over 10 agent trajectories. This resolves both (1) and (2). We find that this metric indicates alignment and does indeed demonstrate the utility of the RoboCLIP alignment score as a reward. Summary below:
\begin{array} {|r|r|} \text{Video || Text} & \text{Robot Opening Green Drawer} & \text{Robot Closing Green Drawer} & \text{Robot Pushing Red Button} & \text{Robot Turning Handle}\\\  \text{Open drawer videos} & \textbf{0.49} & 0.45 & 0.42 & 0.33 \\\  \text{Drawer Close videos} & 0.07 & \textbf{0.1} & -0.74 & -0.84 \\\  \text{Button Press Videos} & -0.6925 & -0.7309	 & \textbf{0.5351} & -0.41 \\\  \text{Faucet Videos} & 	-0.77 & -0.71	 & 0.81 & \textbf{0.84} \\   \end{array}


Thank you once again for your thoughtful reviews! The rebuttal doc is attached below.

---

### Decision · Program_Chairs · 2023-09-21

**Decision:**

Accept (poster)

**Comment:**

The paper proposes a valuable video demonstration-based approach to reward learning for robotic manipulation scenarios, and adjustments to this work made during the rebuttal stage have appreciably improved the submission's empirical evaluation. Thus, this work is recommended for acceptance.